# Artificial Intelligence-Aided Endoscopy and Colorectal Cancer Screening

**DOI:** 10.3390/diagnostics13061102

**Published:** 2023-03-14

**Authors:** Marco Spadaccini, Davide Massimi, Yuichi Mori, Ludovico Alfarone, Alessandro Fugazza, Roberta Maselli, Prateek Sharma, Antonio Facciorusso, Cesare Hassan, Alessandro Repici

**Affiliations:** 1Department of Biomedical Sciences, Humanitas University, 20090 Rozzano, Italy; 2Endoscopy Unit, Humanitas Clinical and Research Center, IRCCS, 20090 Rozzano, Italy; 3Clinical Effectiveness Research Group, Institute of Health and Society, Faculty of Medicine, University of Oslo, 0315 Oslo, Norway; 4Digestive Disease Center, Showa University Northern Yokohama Hospital, Yokohama 224-0032, Japan; 5Division of Gastroenterology and Hepatology, Veterans Affairs Medical Center, Kansas City, MO 64128, USA

**Keywords:** cancer, screening, colonoscopy, technology, innovation

## Abstract

Colorectal cancer (CRC) is the third most common cancer worldwide, with the highest incidence reported in high-income countries. However, because of the slow progression of neoplastic precursors, along with the opportunity for their endoscopic detection and resection, a well-designed endoscopic screening program is expected to strongly decrease colorectal cancer incidence and mortality. In this regard, quality of colonoscopy has been clearly related with the risk of post-colonoscopy colorectal cancer. Recently, the development of artificial intelligence (AI) applications in the medical field has been growing in interest. Through machine learning processes, and, more recently, deep learning, if a very high numbers of learning samples are available, AI systems may automatically extract specific features from endoscopic images/videos without human intervention, helping the endoscopists in different aspects of their daily practice. The aim of this review is to summarize the current knowledge on AI-aided endoscopy, and to outline its potential role in colorectal cancer prevention.

## 1. Introduction

Colorectal cancer (CRC) is the third most common cancer worldwide [1,2,3], with the highest incidence reported in high-income countries in Europe, North America and Oceania [4]. Although the overall prognosis has been improving in the last decades, with a 5-year survival rate of almost 65% [4], colorectal cancer still ranks as the second most common cause of cancer-related mortality [1,2,3]. However, because of the slow progression of neoplastic precursors (i.e., adenomas and serrated lesions) [5], along with the opportunity for their endoscopic detection and resection [6,7], a well-designed endoscopic screening program is expected to strongly decrease colorectal cancer incidence and mortality [7].

In this regard, quality of colonoscopy has been clearly related with the risk of post-colonoscopy colorectal cancer. The Adenoma Detection Rate (ADR), defined as the percentage of patients undergoing colonoscopy with at least one adenoma resected, is currently considered as the most relevant quality indicator correlating with the incidence of interval colorectal cancer (defined as a colorectal cancer diagnosed within 60 months following a negative colonoscopy screening). In detail, a suboptimal endoscopist Adenoma Detection Rate has been associated with a substantial increase in the risk of post-colonoscopy colorectal cancer incidence, whereas an Adenoma Detection Rate increase was effective in reversing such a detrimental effect, offering additional prevention to the patients [8,9,10,11,12].

Nevertheless, a recent systematic review with meta-analysis reported that about 25% of colorectal neoplasms are missed at screening colonoscopy, resulting in an unacceptable variability in the key quality indicator among endoscopists [13]. The most plausible reasons for such variability are both the inability to detect colonic adenomas due to their subtle appearance, especially for small and flat lesions, and the incomplete inspection of the colorectal mucosa due to blind areas at the flexures or behind folds. In particular, unrecognised lesions within the visual field are considered as the most important issue, as several proof-of-concept studies showed an increased detection when a second observer (i.e., nurse or trainee) assisted the first operator [14,15,16]. Innovations such as virtual and dye-spray chromoendoscopy may enhance mucosal contrast, helping to improve the Adenoma Detection Rate; however, all these strategies are operator-dependent and require a learning curve. As a matter of fact, apart from the advent of high definition (HD) imaging, none of such innovations has revolutionized endoscopists’ daily practice.

Recently, interest in the development of artificial intelligence (AI) applications in the medical field has been growing. Through machine learning processes, and, more recently, deep learning, if a very high number of learning samples are available, artificial intelligence systems may automatically extract specific features from endoscopic images/videos without human intervention [17,18,19], helping the endoscopists in different aspects of their daily practice.

The aim of this review is to summarize the current knowledge on artificial intelligence-aided endoscopy, and to outline its potential role in colorectal cancer prevention.

## 2. Principles of Artificial Intelligence

Artificial intelligence is an evolution of general software systems generating an output from an input through an algorithm [17]. The ability of the artificial intelligence software to develop the most adequate algorithm for a specific aim, after an appropriate training, is called Machine Learning (ML). In the endoscopic field, two principal ML methods have been used: handcrafted knowledge and deep learning. The first one represents the first attempt to develop artificial intelligence, characterized by a preset of rules describing knowledge in a well-defined field. Unfortunately, this system doesn’t have the ability to dynamically improve its algorithm (“to learn”), showing poor ability in conditions of uncertainty. Deep learning can be considered as an evolution of handcrafted knowledge. In particular, this method utilizes a convolutional neural network (CNN) that automatically extracts specific features from data without human intervention after an adequate training with a very high number of learning samples. These large artificial convolutional neural networks enhance the ability to “think” and “learn” and improve performance proportionally to the depth of the network. For this goal, deep learning sets neural connections in motion in order to improve its performance through continuous learning. This network is characterized by a layer of input nodes, connected by numerous internal nodes, organized in different levels (Figure 1).

## 3. Artificial Intelligence in Colonoscopy

Computer-Aided Diagnosis (CAD) systems are expected to have at least two major roles in endoscopy practice, mitigating inherent human errors accompanying colonoscopy procedures. Computer-Aided Detection (CADe) systems have the potential to decrease the polyp miss rate [20], contributing to improved adenoma detection [21] (Figure 2).

Computer-Aided Characterization (CADx) systems may improve the accuracy of colorectal polyp optical diagnosis [22], leading to the reduction of unnecessary removal of non-neoplastic lesions, potential implementation of a resect-and-discard and leave-in-situ strategy, and the proper application of advanced resection techniques (Figure 3).

Moreover, the possible role of artificial intelligence systems as a computer-aided quality assurance during colonoscopy (i.e., cecum detector; speedometer; blind spot detector; bowel preparation identifier) is taking place in the artificial intelligence research field.

### 3.1. Artificial Intelligence-Assisted Detection (CADe)

Initial experiences with Computer-Aided Detection systems were reported in the early 2000s [23,24,25]. However, those systems were designed with handcrafted algorithms, limited in real practice application due to both low specificity and low sensitivity, coupled with long processing times. Recently, deep learning-based algorithms have been developed aiming to assist in adenoma detection. After showing promising results in ex vivo studies [26,27,28], the use of artificial intelligence algorithms has been claimed to improve detection of colorectal lesions in several randomized controlled trials [29,30,31,32,33,34,35,36,37,38,39,40,41] (Table 1).

In a recent standard pairwise meta-analysis that pooled data of such randomized trials, artificial intelligence showed a significantly higher detection performance compared to standard colonoscopy, independently from adenoma location, size and morphology [21], providing a 44% Adenoma Detection Rate increase. Moreover, artificial intelligence systems ranked first in an Adenoma Detection Rate increase, showing a significant improvement when compared to other advanced endoscopy techniques (i.e., chromoendoscopy, systems that increase mucosal visualisation) in a network meta-analysis [42].

In more recent randomized tandem trials [20,43,44,45], Computer-Aided Detection colonoscopy reduced the overall adenoma miss rate when compared to standard colonoscopy (Table 2).

However, it is important to underline that the improvement in detection due to AI systems is only related to the increased capacity for detecting lesions within the visual field, that is, dependent on the amount of mucosa exposed by the endoscopist during the scope withdrawal. To tackle this issue, different add-on devices (e.g., Endocuff Vision) have been developed with the aim of improving mucosal exposure [46]. These tools combined with Computer-Aided Detection systems may theoretically maximize the Computer-Aided Detection performances. A proof-of-concept randomized controlled trial is currently assessing the additional diagnostic yield obtained from Endocuff Vision + artificial intelligence-assisted colonoscopy when compared with the yield obtained with artificial intelligence-assisted colonoscopy alone (NCT04676308) [47].

A Computer-Aided Detection system should be thus only considered as a tool for the endoscopist, since his/her skills and experience remain fundamental in order to perform high-quality colonoscopy, and thus to take advantage of the AI support [48,49].

However, it may be argued that a further increase the performance in adenoma detection of expert endoscopists, such as the ones involved in the first trials [29,30,31,32,33,34,35,36,37,38,39,40,41], may provide only little additional benefit.

In this regard, Repici and colleagues [35] recently investigated the role of a novel artificial intelligence Computer-Aided Detection system (GI-Genius, Medtronic Corp., USA) among non-expert endoscopists, reporting promising results, thus showing the opportunity to reduce once and for all the unwanted variation among endoscopists who are involved in screening programs. With this in mind, when applied to large-scale screening programs, AI systems were modelled to avoid colorectal cancer cases and deaths among the screening population [50].

In addition, a recent population-based cohort [51] study on a Dutch fecal immunochemical testing (FIT)-based colorectal cancer screening program showed that the Adenoma Detection Rate of endoscopists is inversely associated with the risk for interval PCCRC (post-colonoscopy colorectal cancers) in FIT-positive colonoscopies. This study showed that for every 1000 patients undergoing colonoscopy, the expected number of interval PCCRC diagnoses after five years was approximately 2 for endoscopists with Adenoma Detection Rates of 70%, compared with more than 2.5, almost 3.5, and more than 4.5 for endoscopists with Adenoma Detection Rates of 65%, 60%, and 55%, respectively. Endoscopists performing colonoscopy in FIT-based screening should thus aim for markedly higher Adenoma Detection Rates compared with primary colonoscopy, and AI could help filling the gap.

High-quality clinical data confirming this model are needed to suggest that AI-assisted endoscopy, by increasing the detection rate and subsequently the Adenoma Detection Rate, may reduce the risk of post-colonoscopy colorectal cancer, and that AI could be particularly helpful in both settings of non-expert endoscopists and FIT-based colorectal cancer screening.

Nevertheless, in recent years, colorectal lesions with a serrated pattern, namely sessile serrated lesions (SSLs) with or without dysplasia, or traditional serrated lesions, have also been identified as precursors of up to 30% of all colorectal cancers [52]. In particular, advanced serrated lesions, defined as either serrated lesions ≥ 10 mm in size, or one of the two serrated polyp subtypes, namely sessile serrated lesions with dysplasia, or traditional serrated lesions, and considered as colorectal lesions that have a high risk of developing into colorectal cancer, have been recently proposed as a possible target for designing an optimal colonoscopy-based colorectal cancer screening program [53].

These lesions are often flat and subtle, being sessile serrated lesions typically located in the proximal colon, and the rarer traditional serrated lesions, often found distally. These features make such lesions difficult to detect, on one hand explaining the wide difference in serrated detection performances among different operators [54], and on the other hand explaining the higher risk of metachronous colorectal cancer after the resection of these lesions when compared with individuals without any significant lesions in baseline colonoscopy [55].

As for colorectal adenomas, the initial experience with Computer-Aided Detection systems showed promising results in improving the detection of serrated lesions, as reported by the comprehensive meta-analysis of Hassan and colleagues [21]. Whereas individual studies did not detect any significant differences in serrated lesions detection performance, pooling their data together increased the diagnostic yield. Then, the benefits of artificial intelligence-assisted colonoscopy appeared clear even in detecting such subtle lesions when compared to standard high-definition colonoscopy.

On the other hand, implementation of AI could increase medical costs in various ways (higher polypectomy, pathological assessment, and surveillance costs, etc.) [56]; however, the savings related to colorectal cancer reduction may mitigate such additional costs [50]. All these models have been thought through considering actual evidence and guidelines in term of endoscopy surveillance after the index colonoscopy [57,58,59]. Even in this case, the relevant impact on adenoma detection due to Computer-Aided Diagnosis systems, and, eventually, the reduction of missed adenomas, may change the paradigm of endoscopy surveillance itself, as we move from planning our strategy based on what we may have left behind to what we have actually detected (and resected). This may allow us to provide a personalized and dynamic AI-driven surveillance strategy for each patient.

### 3.2. Artificial Intelligence-Assisted Characterization (CADx)

Although the benefits of increasing the Adenoma Detection Rate are well known, there is also risk of overdiagnosis as well. As a matter of fact, most of the polyps detected during colonoscopy are diminutive in size (<5 mm) and of low malignant potential. Furthermore, in AI-aided endoscopy, this proportion may be higher, considering that Computer-Aided Detection systems would particularly improve the detection of such subtle lesions [21,29,30,31,32,33,34,35]. Among specifically trained endoscopists, the accuracy in characterizing these small lesions is reported to be <80% [60], with even lower performances reported for non-expert endoscopists. Thus, the diffusion of Computer-Aided Detection systems would lead to a consistent increase in the resection of hyperplastic polyps, which almost never evolve in a malignant lesion. The observation and/or the resection of these polyps would represent an unnecessary waste of time, and an unnecessary financial burden. In this regard, CADx systems may improve the accuracy of colorectal polyp optical diagnosis, leading to the reduction of unnecessary removal of hyperplastic polyps, with the potential implementation of resect-and-discard (when polyps are resected and discarded without histological evaluation) and leave-in-situ (when non-neoplastic lesions located in the rectum and sigmoid are left in situ without resection, as they have no malignant potential) strategies.

The Preservation and Incorporation of Valuable Endoscopic Innovations (PIVI) performance thresholds have been outlined by the American Society for Gastrointestinal Endoscopy (ASGE) [61]. To fulfil the ASGE PIVI thresholds, such a strategy should achieve (1) >90% agreement in assignment of post-polypectomy surveillance intervals compared to pathology-based decisions, and (2) a negative predictive value (NPV) > 90% when used with high confidence.

More recently, the European Society of Gastrointestinal Endoscopy (ESGE) also published an official Position Statement aiming to define simple, safe, and easy-to-measure competence standards for endoscopists and artificial intelligence systems performing optical diagnosis of diminutive colorectal polyps [62]. Thus, according to a panel of experts, in order to implement the leave-in-situ strategy for diminutive colorectal lesions (1–5 mm), it is clinically acceptable if, during real-time colonoscopy, at least 90% sensitivity and 80% specificity is achieved for high-confidence endoscopic characterization of colorectal neoplasia of diminutive polyps in the rectosigmoid. Considering the resect-and-discard strategy, it is clinically acceptable if, during real-time colonoscopy, at least 80% sensitivity and 80% specificity is achieved for high-confidence endoscopic characterization of colorectal neoplasia of diminutive lesions.

The initial CADx systems were developed for magnifying scopes using narrow-band imaging (NBI) based on support vector machine (SVM) models [63,64,65]. More recently, deep learning algorithms based on convolutional neural networks have been used for CADx systems as well, resulting in both higher diagnostic accuracy and faster processing time. An important advantage of such algorithms was the ability to use raw and unprocessed videos, therefore allowing a real-time integration of the AI system during endoscopy.

Mori et al. [66] assessed the performance of a real-time CADx system providing microvascular and cellular visualization of colorectal polyps after application of NBI and methylene blue staining, with high accuracy; in this case, the system could meet required thresholds.

In order to improve the reproducibility in a real-life clinical scenario, Byrne et al. used white light imaging (WLI) and non-magnified NBI videos to test their CADx system [67]. With sensitivity, specificity, PPV, NPV, and accuracy for differentiating diminutive adenomas and hyperplastic polyps of 98%, 83%, 90%, 97%, and 94%, respectively, this resulted in the first CADx study to reach the ASGE PIVI thresholds with a not-magnified NBI setting in real-time clinical practice. Further, equal prediction performance across two different imaging technologies (NBI and Blue Light Imaging (BLI)) has been reported. It is of note that the BLI was not part of the training set [68], suggesting the potential to have a CADx trained and used with different technologies.

More recently, two CADx systems were shown to exceed the PIVI thresholds for implementation of cost-saving strategies when applied to unmagnified imaging in clinical practice [69,70,71].

Further studies have also evaluated the benefit of integrating CADx systems in clinical practice [72,73], underlying the role of AI systems in significantly improving the performance of all endoscopists, with the greatest benefit achieved in trainees (from 63.8–72% to 82.7–84.2%, *p* > 0.001), thus minimizing the differences with expert endoscopists [72]. Interestingly, these results may lead to a relevant cost saving [74].

However, so far only conflictual and not definitive data are available for most of the commercially available AI tools for characterization [75]. On top of that, for reliable data, the performance of CADx systems should be further evaluated in prospective randomized controlled trials, conducted among both expert endoscopists and trainees in academic and community centers.

### 3.3. Artificial Intelligence-Assisted Quality Control

Although most of the studies on AI-assisted endoscopy have focused on adenoma detection and characterization, improving the quality of the endoscopic examination process itself has been recently becoming a more and more travelled path. In this regard, because of the established correlation between the quality of bowel preparation and Adenoma Detection Rate, along with a certain inter-observer variability in quality assessment, different convolutional neural network-based systems have been developed in order to obtain a standardized report [76,77], achieving a rate of accuracy higher than 90%, thus outperforming the endoscopists consulted for the study, according to Zhou et al. [76]. The same system (ENDOANGEL) was also tested for the monitoring of cecal intubation, withdrawal time and exit speed, alerting the endoscopist in case of blind spots caused due to scope sliding [32]. Thus far, automated blind spots identification systems have been initially investigated in upper GI endoscopy [78,79]; however, it may be even more relevant as a quality metric for colonoscopy, as preliminarily reported by McGill et al. [80].

Polyp size measurements are important for treatment selection and the establishment of monitoring intervals; however, currently used subjective methods are flawed. Indeed, Requa et al. [81] developed a highly accurate convolutional neural network to estimate the size of polyps in colonoscopy, dividing them into three size-based groups of ≤5, 6–9, and ≥10 mm (model accuracy: 97%, 97%, and 98%, respectively). More recently, Abdelrahim et al. [82] developed a deep learning model based on convolutional neural networks (CNN) with an 80% accuracy in real-time polyp sizing. Such systems when integrated to CADx could improve polyp management strategies.

## 4. Limitations

Artificial intelligence is a strong focus of interest for health care. In particular, diagnostic endoscopy is an attractive field for the definitive affirmation of AI as an adjunct weapon against colorectal cancer, with a real potential to improve patient care, principally through standardisation of endoscopy quality. As discussed, the integration of Computer-Aided Diagnosis modules in real-time colonoscopy has shown promising results in increasing adenoma detection while improving optical biopsy accuracy. Yet, several issues need to be addressed before AI can be successfully implemented in daily practice.

First, in considering data coming from RCTs supporting the efficacy of Computer-Aided Detection systems in improving the Adenoma Detection Rate, it could be argued that the fact that all but two of the AI studies were performed in China mitigates the generalizability of the results. In this regard, the promising magnitude of the Adenoma Detection Rate increase may be somewhat facilitated by the low baseline Adenoma Detection Rate across the control groups of the Chinese studies. This is to be related with the relatively young age of the enrolled patients; however, the possible impact of a suboptimal endoscopic quality in the first place cannot be excluded. However, the benefit observed in the two Western trials [30,35], with much higher baseline Adenoma Detection Rate values, seems to exclude such a possibility, as the gradient of increase is very similar to what is estimated in Eastern studies [29,31,32,33,34] and meta-analysis [21]. On the other hand, further studies conducted among both Eastern and Western endoscopists with different endoscopic skills would be of paramount importance in order to corroborate present data.

Secondly, because Computer-Aided Detection systems are based on deep learning architectures in which there is no human control on the final algorithm, their outcomes incorporate some unpredictability, and the endoscopist may be presented with false positive (FP) images which should be never considered as suspicious [83,84,85]. Such FPs may potentially hamper the efficiency of Computer-Aided Detection-assisted colonoscopy due to increased withdrawal time, unneeded polypectomies, and higher endoscopist’s fatigue and distraction [86,87,88]. However, despite a relevant number of FPs per colonoscopy (27.3 ± 13.1), only a minority of them (5.7%) required additional exploration, resulting in a negligible 1% increase in the total withdrawal time [89] in a post hoc analysis of an RCT [30], and confirmed by a following comparative study [90] (Figure 4).

Moreover, considering the lack of high-quality clinical data on AI-assisted optical biopsy, further studies would be even more relevant to confirm the accuracy of the different CADx systems, showing how they might impact both endoscopist behaviour and costs in real life through the implementation of “resect and discard” and/or “leave in situ” strategies. As a matter of fact, one of the major concerns about AI-assisted colonoscopy is the over-expectation of the benefits of Computer-Aided Detection in cancer prevention. Although promising, we must not be satisfied with present evidences. Only a properly powered RCT may confirm the benefit in term of cancer incidence and death.

## 5. Conclusions

In conclusion, the burden of missed preneoplastic lesions during colonoscopy is far from being negligible, resulting in still unacceptable cases of colorectal cancer. Artificial intelligence is set to go on stage with regard to different health care fields, and endoscopy may be a fertile terrain for its development and refinement [91,92]. Further studies have the task of both confirming the promising results of first experiences and redesigning the optimal screening setting considering the new technological opportunity. Regarding Computer-Aided Detection systems, to improve their reliability and minimize bias, the performance of Computer-Aided Diagnosis systems should be corroborated in prospective RCTs, conducted in both community and academic centers, and among endoscopists with different levels of experience. Then, the actual impact of adenoma detection improvement on colorectal cancer prevention needs to be assessed in large-scale population-based studies. Preliminary data on both CADx and AI-assisted quality-control systems are promising; however, the lack of high-quality clinical studies prevents any reliable conclusion. Thus, such studies should be considered as a priority in future research agendas. The steps to be done are still many, but the path does not seem so steep anymore.

## Figures and Tables

**Figure 1 diagnostics-13-01102-f001:**
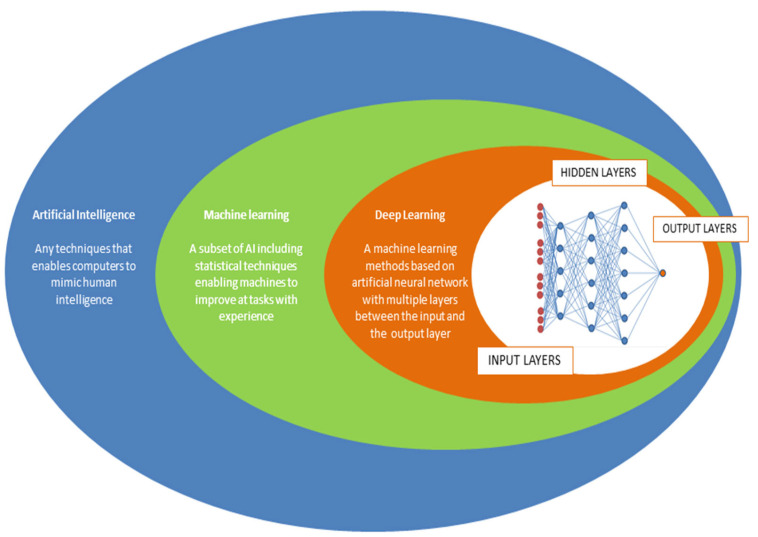
Neural networks and subsets of artificial intelligence.

**Figure 2 diagnostics-13-01102-f002:**
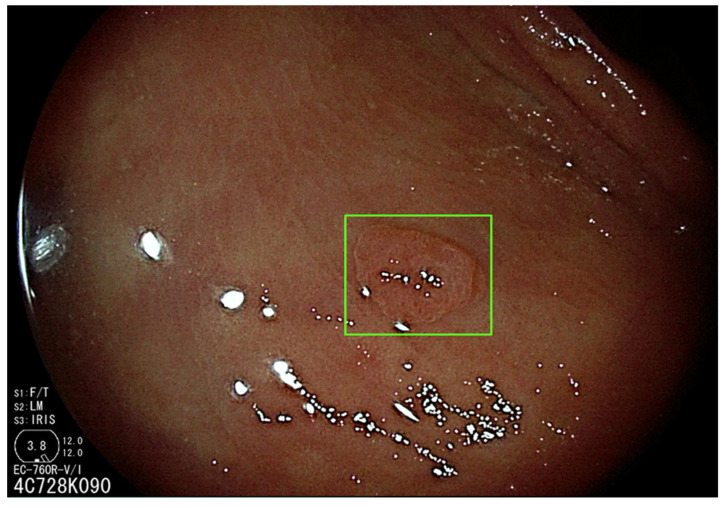
Application of a Computer-Aided Detection (CADe) system (GI-Genius, Medtronic Corp, Minneapolis, MN, USA).

**Figure 3 diagnostics-13-01102-f003:**
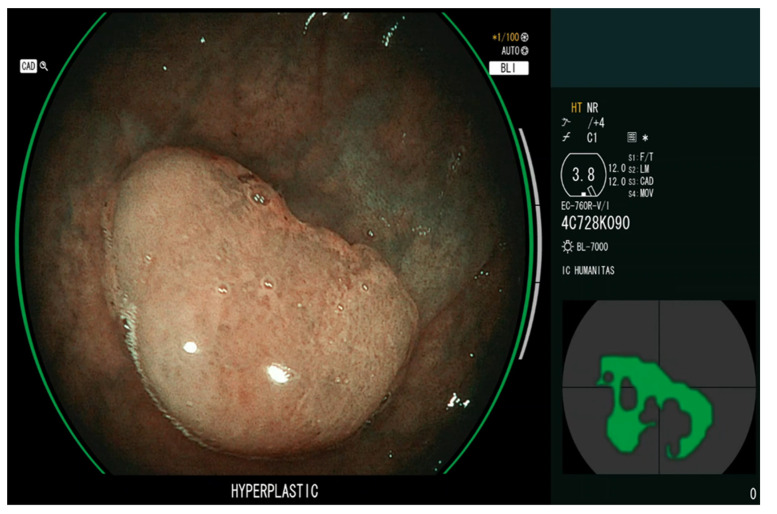
Application of a Computer-Aided Characterization (CADx) system (CAD-EYE, Fujifilm, Japan).

**Figure 4 diagnostics-13-01102-f004:**
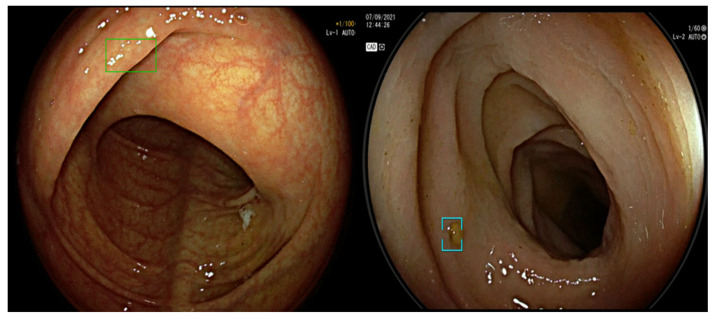
Example of false positive activation of two different Computer-Aided Detection systems due to colonic fold and stool residue.

**Table 1 diagnostics-13-01102-t001:** Parallel Randomized Controlled Trials.

Publication	Country	Study Type	Number of Patients			Gender		Mean Age	Indication		Adenoma Detection Rate (%)		AI System
			Total	AI-Assisted Colonoscopy	Standard Colonoscopy	Men	Women		Screening	Non-Screening	AI-Assisted Colonoscopy	Standard Colonoscopy	
Wang, P. 2019	China	Single Center	1058	522	536	512	546	50	84	974	29	20	EndoScreener
Wang, P. 2020	China	Single Center	962	484	478	495	467	49	158	804	34	28	EndoScreener
Liu, P. 2020	China	Single Center	1026	508	518	551	475	50	66	960	39	24	EndoScreener
Gong, D. 2020	China	Single Center	704	355	349	345	359	50	36	668	16	8	ENDOANGEL
Repici, A. 2020	Italy	Multi center	685	341	344	336	348	61	524	161	55	40	GI GENIUS Medtronic
Su, J.-R. 2020	China	Single Center	623	315	308	307	316	51	216	407	29	17	Self-Developed
Repici, A. 2021	Italy	Multi center	660	330	330	330	330	62	245	415	53	45	GI GENIUS Medtronic
Yao, L. 2021	China	Single Center	539	268	271	235	304	51	534	5	21	15	ENDOANGEL
Liu, P. 2021	China	Single Center	790	393	397	374	416	49	182	608	29	19	EndoScreener
Rondonotti, E. 2022	Italy	Multi center	800	405	395	409	391	61	800	0	53	45	CAD EYE FujiFilm
Shaukat, A. 2022	USA	Multi center	1359	682	677	723	636	60	1359	0	48	44	SKOUT device
Aniwan, S. 2022	Thailand	Single Center	622	312	310	266	356	62	\	\	52	42	CAD EYE FujiFilm
Gimeno-Garcia, A.Z. 2022	Spain	Single Center	312	155	157	165	147	64	206	106	57	45	ENDO AID Olympus

**Table 2 diagnostics-13-01102-t002:** Tandem Randomized Controlled Trials.

Publication	Country	Study Type	Number of Patients			Gender		Mean Age	Indication		Adenoma Detection Rate (%)		AI System
			Total	AI-Assisted Colonoscopy	Standard Colonoscopy	Men	Women		Screening	Non-Screening	AI-Assisted Colonoscopy	Standard Colonoscopy	
Wang, P. 2020	China	Single Center	369	184	185	179	190	47	113	256	35	26	EndoScreener
Kamba, S. 2021	Japan	Multi center	295	178	117	272	23	62	228	67	65	53	Self-Developed
Glissen Brown, J.R. 2021	USA	Multi Center	223	113	110	122	101	61	133	90	50	44	EndoScreener
Wallace, M.B. 2022	Italy	Multi center	230	116	114	157	73	64	230	0	62	61	GI GENIUS Medtronic

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
