# Peer review of "Artificial Intelligence-Aided Endoscopy and Colorectal Cancer Screening"

_diagnostics, 2023, doi:10.3390/diagnostics13061102_

Round 1

Reviewer 1 Report

This is a very interesting paper, with a critical review of Artificial Intelligence-aided endoscopy and Colorectal Cancer screening. Authors provided a critical review of this topic and a deep explanation of AI in the evaluation of endoscopy cancer screening. 

Author Response

Thanks for the support

Reviewer 2 Report

Dear Editor

 This is a well written manuscript regarding current status of AI in colonoscopy. Only minor suggestion.

1. For Table 1 and 2, authors may add the type of study, i.e. single center vs multi center study

2. Table 2, Kamba S , the AI system YOLO v3 should be a type of CNN algorithm, not the name of AI system, please check it

Author Response

 This is a well written manuscript regarding current status of AI in colonoscopy. Only minor suggestion.

Authors (A): Thanks for the support

  1. For Table 1 and 2, authors may add the type of study, i.e. single center vs multi center study

A: Thanks for the suggestion. We added accordingly.

2. Table 2, Kamba S , the AI system YOLO v3 should be a type of CNN algorithm, not the name of AI system, please check it

A: Thanks for highlight this point. We corrected accordingly.

Reviewer 3 Report

The authors present a systematic review of colorectal cancer and artificial intelligence diagnostic tools. It looks as an introduction to a PhD thesis with the topic outlined in lines 293-4. It is a systematic review that does not comply with established standards. It is a list of described papers without key performance indicators such as accuracy, negative predictive value etc., presented in Table 1 and 2. The review brings no additional value with a discussion of presented data. Complying with reporting standards will enhance the quality of the review and may make it a meta-analysis.

I recommend rejection of the article with encouragement of resubmission.

Major.

1.       Please add “systematic” to the title of the manuscript. You compare your report with Ref. 13, which is of the same type.

2.       This is a systematic review. Please add as supporting material a filled form of fulfilling the guidelines of such a review with page numbers and/or sections and paragraphs where the item is described: https://prisma-statement.org/ . PRISMA is for any systematic review. For Artificial Intelligence there are several new extensions, and I suggest the use of DECIDE-AI (https://www.bmj.com/content/377/bmj-2022-070904), however if the authors find some other recommendation, it is also acceptable.

3.       Table 1 does not include a standard parameter of accuracy used in AI evaluation. Please include it. It may be substituted by AUC (area under the curve) value for the model if this is known.

4.       I miss more examples of positively detected lesions and some falsely positive detected ones. Please include some example images.

5.       Line 249 speaks about 90% negative predictive value of AI colonoscopy systems and 90% concordance. These are very important requirements, but none is listed in Table 1 or 2.

6.       The authors may not be aware of MedMNIST database (https://medmnist.com/, https://doi.org/10.1038/s41597-022-01721-8 and dataset in Zenodo https://doi.org/10.5281/zenodo.6496656), where there is PathMNIST database of Colon Pathology with more than 107 thousand images for AI/ML/DL training, but not for clinical use.

Minor.

1.       Please elaborate the captions of Figures 2 and 3. They are the same, while the content is different.

2.       Please include reference number in format [29] in first column of Table 1.

3.       The one-before-last row of Table 1 contains Japanese yen ¥ symbol without description. Please either correct it or explain the symbol.

4.       Please use consistent accuracy measure, either as a fraction of one (0.97 line 316) or percentage (97% lines 277, 318).

5.       What units are in value of 27.3 in line 348? Are these % values?

Author Response

The authors present a systematic review of colorectal cancer and artificial intelligence diagnostic tools. It looks as an introduction to a PhD thesis with the topic outlined in lines 293-4. It is a systematic review that does not comply with established standards. It is a list of described papers without key performance indicators such as accuracy, negative predictive value etc., presented in Table 1 and 2. The review brings no additional value with a discussion of presented data. Complying with reporting standards will enhance the quality of the review and may make it a meta-analysis.

I recommend rejection of the article with encouragement of resubmission.

Authors: Thanks to the reviewer for the comment, however we need to disclose a preliminary misunderstanding. We were invited by the editor to write a NARRATIVE review on the field of AI applied to colorectal cancer prevention. This is not and was never meant to be a SYSTEMATIC review.

Major.

1. Please add “systematic” to the title of the manuscript. You compare your report with Ref. 13, which is of the same type.

A: The title is deliberately generic “Artificial Intelligence-aided endoscopy and Colorectal Cancer screening” since it is not a systematic review. As a matter of fact we have never compared our review with reference 13 (that is a MA and SR on a different topic).

2. This is a systematic review. Please add as supporting material a filled form of fulfilling the guidelines of such a review with page numbers and/or sections and paragraphs where the item is described: https://prisma-statement.org/ . PRISMA is for any systematic review. For Artificial Intelligence there are several new extensions, and I suggest the use of DECIDE-AI (https://www.bmj.com/content/377/bmj-2022-070904), however if the authors find some other recommendation, it is also acceptable.

A: As already stated this is not and was never meant to be a SYSTEMATIC review. 

3. Table 1 does not include a standard parameter of accuracy used in AI evaluation. Please include it. It may be substituted by AUC (area under the curve) value for the model if this is known.

A: Thanks for the comment, however, as the reviewer surely knows, CADe system are not evaluated in clinical studies through accuracy parameters (sensitivity, specificity, ... and AUC value). This is only possible in video-based studies in which the actual number of polyps shown to the AI system is known. This is of course not possible during actual colonoscopies in real patients. And this is why you may find accuracy parameter only in standalone pre-clinical video-based studies.  

On the other hand CADx systems performances may be assessed using accuracy measures using the histology as the ground truth.

4. I miss more examples of positively detected lesions and some falsely positive detected ones. Please include some example images.

A: Thanks for the suggestion. We added some more examples. 

5. Line 249 speaks about 90% negative predictive value of AI colonoscopy systems and 90% concordance. These are very important requirements, but none is listed in Table 1 or 2.

A. Thanks for raising this point, however, as already stated, reporting Table 1 and 2 data on clinical studies on CADe systems (for detection) is not possible to have any accuracy data. As a matter of fact is not a case if data reported in line 249 are referred to characterization performances (then CADx system). 

6. The authors may not be aware of MedMNIST database (https://medmnist.com/, https://doi.org/10.1038/s41597-022-01721-8 and dataset in Zenodo https://doi.org/10.5281/zenodo.6496656), where there is PathMNIST database of Colon Pathology with more than 107 thousand images for AI/ML/DL training, but not for clinical use.

A. Thanks for giving this information, however authors are well aware of these (among lots of others) database, however considering the mostly clinical focus how our review we find reporting them completely off topic.

 Minor.

1. Please elaborate the captions of Figures 2 and 3. They are the same, while the content is different.

A. Thanks for the suggestion, we did it accordingly.

2. Please include reference number in format [29] in first column of Table 1.

A. Thanks for the suggestion, we did it accordingly.

3. The one-before-last row of Table 1 contains Japanese yen ¥ symbol without description. Please either correct it or explain the symbol.

A. Thanks for the comment, we corrected it accordingly.

4. Please use consistent accuracy measure, either as a fraction of one (0.97 line 316) or percentage (97% lines 277, 318).

A. Thanks for the suggestion, we did it accordingly.

5. What units are in value of 27.3 in line 348? Are these % values?

A. Thanks for the comment, however ther is no unit; it is in fact the mean number of false positive activation per colonoscopy, as reported in the text "However, despite a relevant number of FP per colonoscopy (27.3 ± 13.1), only a..." 

Reviewer 4 Report

The literature has not been studied in depth.

It is not reported from which databases or with which Search Strategy the literature search was conducted.

Study Selection details are not provided.

The title and the content are incompatible.

In the study, Computer-aided diagnosis (CAD) systems containing AI, not AI application, were examined.

Scientific contribution is low.

Author Response

The literature has not been studied in depth.

It is not reported from which databases or with which Search Strategy the literature search was conducted.

Study Selection details are not provided.

The title and the content are incompatible.

In the study, Computer-aided diagnosis (CAD) systems containing AI, not AI application, were examined.

Scientific contribution is low.

Authors: We thanks the reviewer for the thoughtful comments, however we need to disclose a preliminary misunderstanding. We were invited by the editor to write a NARRATIVE review on the field of AI applied to colorectal cancer prevention. This is not and was never meant to be a SYSTEMATIC review. Thus, there is no need of both search or study selection strategies. However, considering some of the authors of this review contributed with a large part of the general knowledge on AI in colonoscopy with both individual studies and recent meta-analysis (with appropriate search strategies), we beg to disagree with the reviewer on the quality of literature search.

Round 2

Reviewer 4 Report

it has been nice work. It is appropriate to publish